# Work Engagement, Work Environment, and Psychological Distress during the COVID-19 Pandemic: A Cross-Sectional Study in Ecuador

**DOI:** 10.3390/healthcare10071330

**Published:** 2022-07-18

**Authors:** Carlos Ruiz-Frutos, Ingrid Adanaqué-Bravo, Mónica Ortega-Moreno, Javier Fagundo-Rivera, Kenny Escobar-Segovia, Cristian Arturo Arias-Ulloa, Juan Gómez-Salgado

**Affiliations:** 1Department of Sociology, Social Work and Public Health, Faculty of Labour Sciences, University of Huelva, 21007 Huelva, Spain; frutos@uhu.es; 2Safety and Health Postgraduate Programme, Universidad Espíritu Santo, Guayaquil 092301, Ecuador; 3Faculty of Engineering in Mechanics and Production Sciences, Escuela Superior Politécnica del Litoral, Guayaquil 090902, Ecuador; kescobar@espol.edu.ec (K.E.-S.); caarias@espol.edu.ec (C.A.A.-U.); 4Department of Economy, University of Huelva, 21007 Huelva, Spain; ortegamo@uhu.es; 5Red Cross Nursing University Centre, University of Sevilla, 41009 Sevilla, Spain; javier.fagundo@cruzroja.es

**Keywords:** COVID-19, Ecuador, mental health, psychological distress, work engagement, work environment

## Abstract

Work environments can interfere with the mental health of workers as generators or reducers of psychological distress. Work engagement is a concept related to quality of life and efficiency at work. The aim of this study was to find the relationship between work environment factors and work engagement among the Ecuadorian general population during the first phase of the COVID-19 pandemic to assess their levels of psychological distress. For this purpose, a cross-sectional, descriptive study using a set of questionnaires was performed. Sociodemographic and work environment data, work engagement (UWES-9 scale) scores, and General Health Questionnaire (GHQ-12) scores were collected. The variables that predicted 70.2% of psychological distress during the first phase of the pandemic were being female, with a low level of vigour (work engagement dimension), being stressed at work, and low job satisfaction. The sample showed an intermediate level of engagement in both the global assessment and the three dimensions, being higher in those without psychological distress. With effective actions on work environment factors, mental health effects may be efficiently prevented, and work engagement may be benefited. Companies can reduce workers’ psychological distress by providing safe and effective means to prevent the risk of contagion; reducing the levels of work conflict, work stress, or workload; and supporting their employees with psychological measures in order to maintain ideal working conditions.

## 1. Introduction

Since the SARS-CoV-2 (COVID-19) pandemic declaration by the World Health Organization on 11 March 2020, the way we live and work has changed rapidly [1,2,3]. In Ecuador, as of 18 April 2022, nearly 865,000 cases of COVID-19 were identified, and more than 35,000 deaths had been confirmed. These statistics placed the country among those with the highest number of cases and deaths per capita in the world [4,5]. Many studies have assessed the influence of the COVID-19 pandemic on the development of psychological distress in the general population and health care workers of Ecuador, evidencing a moderate-high level of psychological distress in both groups [6,7,8,9,10], in agreement with other studies in nearby Latin American countries [11,12,13,14].

The first preventive measures in the work environment adopted by agencies and institutions to stop the harmful effects of the pandemic on workers’ health was social distancing [15], since this measure was found to be effective in mitigating the spread of other infectious diseases such as influenza [16]. Following this, the Occupational Safety and Health Administration (OSHA) guidance on workplace preparedness suggested minimising movement and contact between workers, clients, and customers in general; transforming face-to-face meetings into virtual conferences; and encouraging teleworking to prevent disease transmission [17]. However, many authors concluded that Ecuadorians, although they perceived the health crisis as severe and felt greater psychological exhaustion than other countries, showed less adherence to health recommendations and reported lower levels of awareness about their relevance at work. This lack of knowledge could explain the high levels of psychological distress found in the Ecuadorian population [7,8]. In fact, a group of authors surveyed health care workers in Ecuador during the first month of the pandemic and revealed that nearly 25% of these professionals believed in conspiracy theories about the virus dissemination, prevalent belief associated with high anxiety and distress [10]. For the United Nations [18], the collateral effects of such preventive measures on people’s mental health has not been sufficiently assessed: Assuming that social distancing and telework have had negative and positive effects on the population and workers, these deserve to be analysed.

In this regard, in Ecuador, studies on the effects of the pandemic on telework according to ergonomics and health effects have been published [19,20] and also regarding the work–family conflicts among the general population [21]. Among the negative effects of teleworking on health, the difficulty establishing a physical and temporal separation between work and family environments has been the main concern, affecting levels of satisfaction and commitment to work (called work engagement). On the other hand, teleworking also has positive effects, such as reduced commuting, which facilitates access to work for people, especially those with mobility deficits. Improvements in productivity and increased engagement have also been observed [19,21,22]. Other international studies on teleworkers have found that the factors that increase engagement are communication with superiors, the reduction of long working hours, and control of adequate sleep hours [23]. 

Following this, engagement is described as the positive and satisfying work-related state of mind expressed through three dimensions: vigour (desire to invest effort in work), dedication (proactive participation), and absorption (concentration during work) [24]. This multi-axial concept is closely related to the organisational environment, work resources, professional needs, work demands, and demographic variables [25]. High levels of work engagement are associated with low burnout and cynicism and with high efficiency [26]. Studies on health care professionals who were working during the COVID-19 lockdown revealed a high level of engagement, with nurses standing out [7,9,10,27]. Additionally, studies about work engagement on Ecuadorian and Peruvian health care workers during the pandemic showed that high professional self-efficacy leads to higher work engagement and in turn to increased quality of life and self-care attitudes among the professionals [9,14]. Meanwhile, in a group of Spanish non-health care workers, an association was observed between a higher level of psychological distress and lower levels of engagement in its three dimensions [28]. 

Despite these assessments of psychological distress and work engagement performed in Ecuadorian health care workers, studies that relate work engagement, work environment, and psychological distress have not been conducted in the Ecuadorian general population. Therefore, the aim of this study was to find the associations between psychological distress, work engagement, and the work environment during the period of COVID-19 confinement in Ecuador. Our hypothesis was that the level of work engagement and the safety provided by the work environment would be associated with the development of psychological distress in Ecuadorian workers. This study contributes to identifying the most influential occupational factors that produce psychological distress in general workers in this pandemic context, so that specific preventive measures can be introduced by health authorities and health care managers in workplaces to reduce the impacts on people’s mental health and increase workers’ work engagement.

## 2. Materials and Methods

### 2.1. Study Design

This was a cross-sectional study that used multiple validated questionnaires: one for sociodemographic data [29], the General Health Questionnaire (GHQ-12) [30], and the Utrecht Work Engagement Scale (UWES-9) [31].

### 2.2. Participants

The estimated sample size, considering the total Ecuadorian population, was 2481, with 95% confidence level, 2.2% precision, and a loss adjustment of 20%. The loss was 12.9%. Thus, the final sample consisted of 2161 participants from the 24 provinces of Ecuador at the end of the survey collection period.

Then, non-probabilistic snowball sampling was used, disseminating the study through social networks and various public institutions.

Participants could access the survey if they met the following criteria: having worked and resided in the country during the first phase of the pandemic, being over 18 years of age, and having agreed to participate voluntarily and anonymously in the study via informed consent.

### 2.3. Measuring Instruments

A questionnaire validated in Spain by a group of experts [29] was used to collect data based on similar studies on other pandemics that obtained a Cronbach’s α coefficient of 0.86 and good psychometric properties. It was then culturally adapted to the population of Ecuador to ensure good understanding of the items and to include country-specific data. The questionnaire included sociodemographic variables such as sex, age, marital status, education level, number of children, pet ownership, and type of work, differentiating between working away from home and teleworking.

Information was also collected on eleven variables related to the work environment during the pandemic (Table 1): the effectiveness of preventive measures, perceived safety, level of labour conflict, risk of infection at work, degree of acceptance of the disease, workload, stress, degree of satisfaction, and need for psychological support for different collectives, measured with a range from 1 (never) to 10 (always).

Psychological distress was measured with the Goldberg scale using the General Health Questionnaire (GHQ-12) [30], a scale designed to assess mental health using 12 questions or items rated on Likert scales from 0 to 4, with an overall score from 0 to 12 points. A positive indication of psychological distress for all individuals was a score greater than or equal to 3 on the GHQ-12 (Cronbach’s α = 0.880).

The Utrecht Work Engagement Scale (UWES-9) (Cronbach’s α = 0.935) [31] was used to assess work engagement. This questionnaire consists of 9 questions with Likert scale answers from 0 (never) to 6 (always), distributed in 3 dimensions: vigour, dedication, and absorption. The total score of the UWES categories is standardized at Very high (≥5.51) [P95, Max]; High (4.67–5.50) [P75, P95]; Intermediate (2.89–4.66) [P25, P75]; Low (1.78–2.88) [P5, P25]; or Very low (≤1.77) [Min, P5]. The internal consistency obtained for the different dimensions was: α = 0.872 for vigour, α = 0.877 for dedication, and α = 0.781 for absorption.

### 2.4. Procedure

Data were collected through the Qualtrics^®^ platform (Qualtrics, Seattle, WA, USA), for the survey and storage and distributed with the support of scientific associations, universities, and institutions. Social media from public institutions were used to invite participation, with information about the study and the link to access the questionnaire. As the questionnaires were not sent individually by the investigators, participants were asked to share the questionnaire with their colleagues and organisations, after completion, through snowball sampling.

The questionnaire could be accessed using different electronic means with internet access (computer, tablet, or mobile phone). Data collection took place during the first phase of the pandemic, between 2 April and 17 May 2020.

A total of 320 questionnaires whose response rate was less than 99% of the items completed were discarded to avoid calculation bias.

### 2.5. Data Analysis

Once the database was cleaned, descriptive measures were determined according to the type of variable. The existence or not of a relationship between the different variables (sociodemographic, UWES dimensions, and work environment) with respect to the presence or absence of psychological distress was assessed using the chi-squared test of association and Student’s t-test for independent samples.

A binary logistic regression analysis allowed for identifying those variables that played a more relevant role with respect to psychological distress. The Kolmogorov–Smirnov test rejected the normality of the variables included in the model; however, the skewness and kurtosis coefficients presented low values, lower than two in absolute value, for all the quantitative variables included in the model. Forward selection was performed by considering the likelihood ratio statistic, estimating odd ratios (ORs). Confidence intervals were provided for this measure of association. In addition, different goodness-of-fit measures were used: the Hosmer–Lemeshow test, percentage of correctly classified values, sensitivity, and specificity. Multivariate normality and multicollinearity issues were taken into account in the data analysis. The independent variables in the model showed moderate collinearity, i.e., minimal correlation between covariates, and therefore were not eliminated.

All analyses were performed with SPSS 26.0 statistical software (IBM, Armonk, NY, USA).

### 2.6. Ethical Considerations

Informed consent was obtained from all study participants prior to the start of the questionnaire. The ethical principles established in the Declaration of Helsinki of 1964 and its newest declaration of Fortaleza in 2013 were followed. The study was authorised in Ecuador by the Ethics Research Committee of the Universidad San Gregorio de Portoviejo (USGP-DI-049-2021) and in Spain by the Ethics Committee of Huelva, belonging to the Ministry of Health of Andalusia, Spain (PI 036/20).

## 3. Results

A total of 2161 participants formed the sample of this study. Among them, 53% were women, and the mean age of more than half of the sample was 32 years or younger (median: 32). Additionally, 54% were unmarried at the time of the survey, although 52% had at least one child. Most, 84%, of the sample had university studies or higher, and there were similar numbers of public (45%) and private employees (40%). About half, 52.8%, of the sample worked from home during the study, and 47.2% worked outside the home.

As can be seen in the GHQ-12 analysis in Table 2, 62.6% of the sample had a high level of psychological distress (PD) (GHQ-12 ≥ 3), M = 4.31 (SD = 3.41).

Observing the associations between the sociodemographic variables and PD, a higher level of PD (with *p* < 0.001) was found among women (OR = 1.908; 95% CI = 1.600–2.276) and among individuals with university studies or higher (OR = 1.524; 95% CI = 1.205–1.926), with no statistically significant difference found with respect to the other variables (age, marital status, number of children, having a pet, organisation they worked for, or telework) (Table 3).

As can be seen in Table 4, the overall mean for work engagement (UWES) was M = 4.5 (SD = 1.2), which is considered an intermediate level of engagement in the UWES-9, being lower among those with PD (M = 4.2; SD = 1.2) than among those without PD (M = 4.9 (SD = 1.0), who reported high engagement with *p* < 0.001. 

This association is also significant with respect to the three dimensions of work engagement: Vigour at work had the lowest mean score (M = 4.2; SD = 1.4) and the lowest score for those with PD (M = 3.9; SD = 1.4), although the effect size and statistical power were higher than others. Dedication received the highest mean score (M = 4.7; SD = 1.3) and also the highest score for those who did not show PD (M = 5.1; SD = 4.4). Absorption stayed within the mean (M = 4.5; SD = 1.2) and received higher scores for those without PD (M = 4.8; SD = 4.4 vs. M = 4.4; SD = 1.2). 

It is possible to observe the UWES scores at the P75 percentile, where the effect sizes are considered high. In this case, absorption was ‘very high’ (5.67) and dedication (5.67) along with vigour (5.33) were ‘high’.

Table 5 shows that it is indeed the need for psychological support that has the highest values, for people and their families affected by the disease (M = 9.2; SD = 1.7), for professionals and volunteers (M = 9.0; SD = 2.0), and also for the general population (M = 8.9; SD = 1.9).

When analysing the possible associations between workplace characteristics in relation to the response to the pandemic and the likelihood of developing PD, different effect sizes occur. The higher effect size was found for the association between stress at work and PD, and job satisfaction and level of labour conflicts were also relevant (Table 5).

There was a higher level of PD in workplaces where an increase in work conflicts occurred (M = 5.7; SD = 3.1 vs. M = 4.8; SD = 3.1); *p* < 0.001; in places where the risk of becoming infected was high (M = 7.1; SD = 3.2 vs. M = 6.8; SD = 3.2); *p* < 0.05; and in environments where higher workload (M = 7.4; SD = 3.0 vs. M = 6.6; SD = 3. 1) and more stress at work were perceived (M = 7.7; SD = 2.7 vs. M = 5.7; SD = 3.2), *p* < 0.001. Likewise, the level of PD was higher among those who thought that psychological support was needed for people and families affected by the disease (M = 9.3; SD = 1.6 vs. M = 9.1; SD = 1.9), *p* < 0.001, as well as among those who thought psychological support was needed for professionals and volunteers (M = 9.1; SD = 1.8 vs. M = 8.8; SD = 2.2), *p* < 0.001. No statistically significant difference was observed with respect to the need for psychological support in the general population (Table 5).

On the contrary, there was a lower level of PD in workplaces where the effectiveness of preventive measures to perform work effectively exist (M = 7.3; SD = 2.7), in workplaces that provided the means to perform work safely (M = 7.3; SD = 2.7), and among those individuals with higher job satisfaction (M = 7.2; SD = 2.4 vs. M = 6.3; SD = 2.5), *p* < 0.001. The degree of acceptance of the disease reached very similar results (M = 5.1; SD = 3.5 vs. M = 4.9; SD = 3.5) (Table 5).

Table 6 shows that the variables that predict psychological distress among Ecuadorian workers, with a percentage of 70.2%, are being a woman (OR = 1.546; 95% CI = 1.273–1.876), with a low level of vigour (OR = 0.874; 95% CI = 0.850–0.900), being stressed at work (OR = 1.190; 95% CI = 1.152–1.230), and low reassurance with their work in the current COVID-19 situation (OR = 0.901; 95% CI = 0.86–50.939).

## 4. Discussion

The present study revealed a number of factors that were associated with psychological distress caused by COVID-19 in the occupational context. Considering, first, the relevance of the perception of stress in the work environment caused by the pandemic, it directly affected the satisfaction of the employee with the organisation and the job position. This may affect the efforts and contributions made by the worker on a daily basis, generating an emotional and physical negative influence that could be perceived as psychological distress. 

Women showed significantly higher levels of depression, anxiety, and stress than males, a common finding in most studies around the world [10,11,12,32,33]. Restrictive measures regarding schools and day-care centres may significantly increase the burden on women at home, leading to fatigue and a reduction in their work performance [12,21,27]. On the other hand, an increase in domestic violence against women during quarantine due to the pandemic and a higher risk of losing their jobs and incomes could be the reasons for our findings. In fact, we found inverse correlations between age and the levels of depression, anxiety, and stress. The reason may be that younger individuals tend to be more concerned about future consequences and the negative impacts of the pandemic on the global economy and job availability [11,34]. Likewise, young people have greater and more continuous access to inaccurate information due to their use of social media, which can affect their mental health [33].

In relation to evidence from previous studies [27,28], work engagement in this study appears in an intermediate level in its global assessment as well as in its three dimensions. Better scores on the UWES-9 are associated with lower levels of PD, confirming the role played especially by the vigour dimension in predicting the level of PD generated during the COVID-19 pandemic. These results could be compared with the findings of a study on Ecuadorian health care workers, whose levels of vigour were high and positively related to age, years of working experience, and salary [9].

Although we also confirmed+ the existence of psychological distress in the studied setting, its level (GHQ ≥ 3 = 62.6%) is not very different from that observed in other countries where the same instrument was used, such as Argentina (60.9%) [13], Spain (65.2%) [35], Peru (59.68%) [11], and Portugal (57.2%) [36], and it was lower than in Chile (78.83%) [12]. This study also found a perceived need for psychological support in professionals and volunteers involved in responding to the pandemic, including those affected by the disease and their families, a fact that has been previously reported in health care workers [37] and non-health care workers [28]. In this regard, the incidence of the disease in Ecuador may have produced an engaged health care force, with high levels of the three dimensions of engagement, a fact that some authors have related to the promotion of social support as a protective factor of psychological well-being [9]. However, another study on the Ecuadorian general population identified an acceptable level of knowledge about COVID-19 but stated that this was not enough to motivate a change of attitude towards the pandemic and how to deal with it [38]. Therefore, the emergence of possible long-term effects on mental health is still to be clarified in the general population; the rapid availability of psychosocial support resources is also needed to prevent long-term negative effects of the pandemic [33,39,40].

Another interesting finding of this study points to the association between workers’ perceived insecurity in their workplaces regarding protecting themselves from possible infection and higher levels of PD, which corroborates previous evidence on the mental health effects of COVID-19 presence in the work environment [41]. Those responsible for protecting workers’ health in companies should take these aspects into account, considering that the immediate availability of preventive contact and respiratory measures could reduce the potential spread of the disease. This is not the only factor to consider in reducing the levels of PD in workers. It has been shown that the level of conflict in the company, the workload, work stress, and the degree of satisfaction with the work are also factors intrinsically associated with mental health in times of pandemic. The perceived safety of the workplace and the effectiveness of the preventive measures given at work have also been identified as the most influential factors in reducing the impacts on mental health [39].

In the present study, no differences were found between the perceived PD of those who teleworked and those who worked away from home, something that has been found in other studies where PD was higher among those who worked away from home [29]. This could be justified by the presence of psychosocial stressors in teleworking [19] and by the risk of contagion during the essential activities of those working away from home [42].

The limitations of this study are related to the non-probabilistic sampling technique used, as well as the inequality between the percentages of participants with and without university studies or higher (84.4% of the sample was composed of people with university studies), which could have affected the results from the perspective of the use of information and shared beliefs. For the correct interpretation of the results, it should be kept in mind that the data were obtained during the first phase of the pandemic, when a strict confinement system was imposed [15]. The differences of this study with other internationally published studies have been considered and can be explained by variations in the percentage of people affected, the type of health system, and the movement restriction measures adopted, as well as whether or not the population analysed were exclusively health workers. It is necessary to remember that this is a cross-sectional study that does not allow establishing a cause–effect relationship.

With effective action on work environment factors, mental health effects may be efficiently prevented, while work engagement may increase. This investigation should encourage those responsible for managing occupational health and safety in companies to reduce the levels of psychological distress among workers by both providing preventive measures to reduce contagion and increasing awareness of the disease and its prevention. Likewise, self-efficacy in the control of stress and workload while maintaining low levels of work-family conflicts would lead to an increase in job satisfaction. Organisations can increase their employees’ work engagement by encouraging virtual meetings, refresher courses, webinars to manage anxiety and stress, or informal conversation sessions, tools that have been found to be effective during the current pandemic and may help even in a post-COVID-19 phase [43]. Similarly, family and peer support, self-efficacy enhancement, and worker resilience have been shown to increase engagement during stressful situations [44]. Specifically, in health care workers involved in the pandemic, psychological support has been found to be a highly effective measure for reducing negative mental health effects [37] and is valid for the general population [45]. Moreover, persistent COVID-19 has led to the development of specific guidelines to minimise its effects at work and help workers recover [46].

This research has been performed within the same research project that is being developed in 18 countries in Latin America and Europe, which will facilitate comparisons in the near future. 

## 5. Conclusions

The presence of psychological distress in the Ecuadorian sample of this study reached 2.6%, which is a similar value to the one found in studies in other Latin American countries like Argentina and Peru.

The factors that, to a large extent (70.2%), predicted the development of PD during the first phase of the COVID-19 pandemic in Ecuador were being a woman and having low levels of the vigour work engagement dimension, high work stress, and low job satisfaction. 

In relation to work engagement, an intermediate level was reached by the sample in its global assessment as well as in its three dimensions. However, individuals with higher levels of engagement had lower levels of psychological distress.

Lastly, in relation to the work environment, most of the sample agreed on the need for psychological support for those who care for patients and their families. Furthermore, most of the participants corroborated the importance of reducing the risk of contagion at work, both with appropriate preventive measures and by increasing the safety of the workplace. Reducing labour–family conflicts, workloads, and work stress could be beneficial for increasing job satisfaction during this pandemic as well.

## Figures and Tables

**Table 1 healthcare-10-01330-t001:** Questions about the work environment in relation to the pandemic.

Variable	Question about Work Environment
Question 1. Effectiveness of preventive measures	Has your department, service or company provided the workers with the necessary means and material to effectively carry out their job?
Question 2. Perceived safety	Has your department, service or company provided the workers with the appropriate means and material to safely carry out their job?
Question 3. Level of labour conflict	Have labour conflicts between partners increased in your workplace during the pandemic?
Question 4. Risk of infection at work	Is there a risk of getting infected at your profession or working environment?
Question 5. Degree of acceptance of the disease	Do you accept the risk of getting infected as part of your job?
Questions 6, 7 and 8. Need for psychological support to professionals, volunteers, patients, families and general population.	Do you believe it would be important to offer psychological support to professionals and volunteers who are actively taking part in the COVID-19 health crisis?
	Do you believe it would be important to offer psychological support to persons and their families who are directly affected by COVID-19?
	Do you believe it would be important to offer psychological support to the general population to deal with the COVID-19 health crisis?
Question 9. Workload	Do you consider there has been an increase in the workload since the health crisis started?
Question 10. Stress	Do you feel stressed about COVID-19 at work?
Question 11. Job satisfaction	Do you feel reassured at your workplace during the present COVID-19 situation?

**Table 2 healthcare-10-01330-t002:** Psychological Distress: General Health Questionnaire GHQ-12.

*N*	2161
Mean (SD)	4.31 (3.41)
Minimum/Maximum	0/12
P25/P50/P75	1/4/7
GHQ ≥ 3	*N* = 1352 (62.6%)
GHQ < 3	*N* = 809 (37.4%)

**Table 3 healthcare-10-01330-t003:** The associations between the sociodemographic variables and psychological distress (*N* = 2161).

	*N* (%)	Psychological Distress by GHQ-12		Statistical	Odds Ratio (Confidence Interval at the 95 Level)
		NO	YES		
Sex					
Male	1015 (47.0)	45.4	54.6	52.069 *	1.908
Female	1146 (53.0)	30.4	69.6		(1.600, 2.276)
Age (median = 32) (*N* = 2143)					0.855 (0.718, 1.019)
32 or younger	1137 (53.1)	35.8	64.2	3.062	
Older than 32	1006 (46.9)	39.5	60.5		
Marital status					1.079 (0.906, 1.285)
With a partner	979 (45.3)	38.4	61.6	0.719	
Without a partner	1182 (54.7)	36.6	63.4		
Educational level					1.524 (1.205, 1.926)
Without university studies	337 (15.6)	46.0	54.0	12.484 *	
University studies or higher	1824 (84.4)	35.9	64.1		
Children					1.127 (0.946, 1.342)
Yes	1127 (52.2)	38.8	61.2	1.803	
No	1034 (47.8)	36.0	64.0		
Pet					1.059 (0.888, 1.262)
Yes	1221 (56.5)	38.2	61.8	0.405	
No	940 (43.5)	36.9	63.1		
Job situation					-
Self-employed	326 (15.1)	39.0	61.0	5.760	
Public employee	974 (45.1)	34.7	65.3		
Working for a private company	861 (39.8)	40.0	62.6		
Teleworking or not					0.923 (0.775, 1.099)
From home	1141 (52.8)	36.5	63.5	0.817	
Away from home	1020 (47.2)	38.4	61.6		

* *p* < 0.001.

**Table 4 healthcare-10-01330-t004:** The associations between the UWES dimensions and psychological distress (*N* = 2161).

UWES Score Categories			Psychological Distress by GHQ-12			
	Min/P25/P50/P75/Max	Mean (SD)	NO M (SD)	YES M (SD)	Statistical	Effect Size
Vigour	0/3.33/4.33/5.33/6	4.2 (1.4)	4.8 (1.1)	3.9 (1.4)	15.960 **	0.64
Dedication	0/4.00/5.00/5.67/6	4.7 (1.3)	5.1 (1.1)	4.4 (1.3)	12.194 **	0.50
Absorption	0/3.67/4.67/5.67/6	4.5 (1.2)	4.8 (1.1)	4.4 (1.2)	8.638 **	0.30
Total	0/3.67/4.77/5.44/6	4.5 (1.2)	4.9 (1.0)	4.2 (1.2)	13.375 **	0.55

M: Mean; SD: Standard Deviation; ** *p* < 0.001.

**Table 5 healthcare-10-01330-t005:** The associations between work environment and psychological distress (*N* = 2161).

		Psychological Distress by GHQ-12			
	M (SD)	NO M (SD)	YES M (SD)	Statistical	Effect Size
Question 1. Effectiveness of preventive measures	6.8 (2.9)	7.3 (2.7)	6.6 (3.0)	5.930 **	0.26
Question 2. Perceived safety	6.9 (2.9)	7.3 (2.7)	6.6 (3.0)	5.396 **	0.23
Question 3. Level of labour conflict	5.3 (3.1)	4.8 (3.1)	5.7 (3.1)	−6.760 **	0.30
Question 4. Risk of infection at work	7.0 (3.2)	6.8 (3.2)	7.1 (3.2)	−2.170 *	0.10
Question 5. Degree of acceptance of the disease	5.0 (3.5)	5.1 (3.5)	4.9 (3.5)	1.790	0.08
Question 6. Need for psychological support (professionals and volunteers)	9.0 (2.0)	8.8 (2.2)	9.1 (1.8)	−3.658 **	0.17
Question 7 Need for psychological support (patients and families)	9.2 (1.7)	9.1 (1.9)	9.3 (1.6)	−2.943 **	0.14
Question 8. Need for psychological support (general population)	8.9 (1.9)	8.8 (2.0)	9.0 (1.9)	−1.949	0.09
Question 9. Workload	7.1 (3.1)	6.6 (3.1)	7.4 (3.0)	−5.595 **	0.25
Question 10. Stress	7.0 (3.1)	5.7 (3.2)	7.7 (2.7)	−14.862 **	0.65
Question 11. Job satisfaction	6.6 (2.5)	7.2 (2.4)	6.3 (2.5)	8.545 **	0.37

* *p* < 0.05; ** *p* < 0.001.

**Table 6 healthcare-10-01330-t006:** Binary logistic regression results for psychological distress (*N* = 2161).

	Odds Ratio (Confidence Interval at the 95% Level)
SEX (ref. Male)	1.546 ** (1.273, 1.876)
UWES: Vigour	0.874 ** (0.850, 0.900)
Question 10. Stress	1.190 ** (1.152, 1.230)
Question 11. Job satisfaction	0.901 ** (0.865, 0.939)
Sensitivity/Specificity	84.2/46.8
Correctly classified percentage	70.2
R^2^	0.235
Hosmer-Lemeshov test	χ^2^ = 2.445 (*p* = 0.964)
Omnibus test	χ^2^ = 407.903 (*p* < 0.001)

** *p* < 0.001.

## Data Availability

Datasets available upon reasonable request to the corresponding author.

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
