# Peer review of "Work Engagement, Work Environment, and Psychological Distress during the COVID-19 Pandemic: A Cross-Sectional Study in Ecuador"

_healthcare, 2022, doi:10.3390/healthcare10071330_

Round 1

Reviewer 1 Report

This is an interesting study that I enjoyed reading. I believe the manuscript needs some changes before it can be accepted for publication. I have reported some comments / suggestions belove:

1) I believe it is important that the authors insert the hypotheses of the study after having indicated the objectives

2) were there any exclusion criteria that the authors applied a priori or a posteriori?

3) I suggest to insert the alpha in the tools section and not in the tables which are already very dense

4) in table 2 it is not clear what “yes/no” refer to

5) I suggest to insert in the discussion a limits section

6) I suggest to improve the conclusions section

Author Response

This is an interesting study that I enjoyed reading. I believe the manuscript needs some changes before it can be accepted for publication. I have reported some comments / suggestions belove:

1) I believe it is important that the authors insert the hypotheses of the study after having indicated the objectives.

The hypothesis of this investigation has been included in the last paragraph of the Introduction section.

2) were there any exclusion criteria that the authors applied a priori or a posteriori?

The exclusion criteria of this study have been described in the Participants and Procedure sections:

  • All questionnaires whose response rate was of less than 99% of the items required were discarded to avoid calculation bias.
  • Only for Ecuadorian population who were working and residing in the country during the first phase of the pandemic.
  • Only those over 18 years of age who agreed to participate voluntarily in the study via an informed consent were admitted.

3) I suggest to insert the alpha in the tools section and not in the tables which are already very dense.

The alpha of each tool has been included in the instruments’ section and has been eliminated from the tables.

4) in table 2 it is not clear what “yes/no” refer to

We have modified the headings in Tables 2 and 3 (now tables 3 and 4) so as to indicate Psychological Distress “yes or no”, measured by the GHQ-12.

5) I suggest to insert in the discussion a limits section

The limitations of this investigation have been included in the Discussion section.

6) I suggest to improve the conclusions section

The conclusions have been revised.

Reviewer 2 Report

The review document is attached.

Author Response

Comments and Suggestions for Authors

Abstract

The abstract must highlight the contribution of the study.

Thank you for your suggestion. The abstract has been revised.

Introduction

The introduction section must offer the study gap, research question and objectives. Also, highlight the contribution of current work.

The introduction has been completely revised to increase its interest and coherence, and also to highlight the study gap, research question, objectives, and contributions of this study.

Method and procedure

No specific information was offered to represent the sample size calculation and information about the target respondents. The sample size calculation must be scientifically performed to achieve the power and effect size. I suggest the authors to use the: Faul, F., Erdfelder, E., Lang, A.-G., & Buchner, A. (2007). G*power 3: A flexible statistical power analysis program for the social, behavioural, and biomedical sciences. Behav. Res. Methods.39, pp. 175–191. Many procedural and statistical tools are available to manage the common method bias (Please see Podsakoff, P.M., Mackenzie, S.B. & Podsakoff, N.P. (2012) Sources of Method Bias in Social Science Research and Recommendations on How to Control It. Annual Review of Psychology. 63 (1): 539-569).

The estimated sample size was 2481, with 95% confidence level, 2.2% precision, and a loss adjustment of 20%. Finally, the loss was 12.9%, leaving a sample size of 2161. This information has been included in the Materials and Methods section.

The authors must report the multivariate normality and multicollinearity issues. Please follow and check Hair, Black, Babin, and Anderson (2014). Multivariate Data Analysis, 7th edition).

The Kolmogorov-Smirnov test rejected the normality of the variables included in the model; however, the skewness and kurtosis coefficients presented low values, less than two in absolute value, in all the quantitative variables included in the model.

However, we would like to report an error detected and corrected in the regression model. The variable Question 6 (identified with the question Do you believe it would be important to offer psychological support to professionals and volunteers who are actively taking part in the COVID-19 health crisis?) showed a strong rejection of normality in the last check. This rejection of normality in the variable made us re-evaluate the logistic regression analysis and detect an error, which we assume to be a transcription error, given that it only affects this variable. All the values of the model are correct, including the validation, except those corresponding to variable Question 6. Need for psychological support (professionals and volunteers), which has been removed from table 6.

Thank you very much for your comment, which allowed us to detect the mistake.

Odds Ratio

(Confidence Interval at the 95% level)

SEX (ref. Male)

1.546** (1.273, 1.876)

UWES_Vigour

.874** (.850, .900)

Question 6. Need for psychological support (professionals and volunteers)

1.074** (1.022, 1.128)

STRESS

1.190** (1.152, 1.230)

SATISFACTION

.901** (.865, .939)

Sensitivity / Specificity

84.2 / 46.8

Correctly classified percentage

70.2

R2

.235

Hosmer-Lemeshov test

χ2=2.445 (p=.964)

Omnibus test

χ2=407.903 (p<.001)

On the other hand, the independent variables of the model present moderate collinearity, i.e. minimal correlation between covariates and therefore were not eliminated.

In the data analysis, it has been included that the analysis of multivariate normality and multicollinearity issues was taken into account.

Results

Authors must offer detailed meaning of Table 3.

Thank you for your remark. Table 3 has been revised and detailed in the main text.

Authors can offer the full meaning of Table 4 and how the different work environments affect psychological distress.

Table 4 has been revised and detailed in the main text.

The meaning of different work environments can be taken to the methods and procedure section.

We have described the variables related to work environment in the Methods section and Table 1.

Discussion

No critical discussion is offered in the present work, and the authors must discuss study findings and compare the current study results with previously published results. A normal suggestion is that the discussion must link with the research question and objectives.

The discussion has been completely revised to focus on the study findings, adding recent and pertinent references to compare, and increasing its visualisation.

Conclusion

The section must discuss the theoretical and managerial implications that may help the practice or contribute to the theory. The study limitation was not fully presented, leading to future research work.

We have revised the conclusions to show all the relevant outcomes and main contributions.

The study limitations have been included in the Discussion section.

The contributions of this study have been presented at the end of the Discussion.

General

A straightforward thought was absent in writing.

The text has been revised to increase its legibility and cohesion.

English language editing is required.

The manuscript has been revised by an English native expert.

Recent references must be used in the study. Following the journal guidelines, the referencing style was not consistently used in the current manuscript.

New references have been included in this review and they have been checked for appropriateness.

Reviewer 3 Report

Strengths: literature review, methodology and sample, analysis of results

Weaknesses: Conclusions

Improvement proposals

Line 83 – “Despite studies on work engagement and the psychological effects of COVID-19 that have been conducted in Latin America, none are available in our immediate environment.” Authors must demonstrate why studies are not readily available.

Line 97 – “A non-probabilistic, snowball sample of 2161 questionnaires was obtained from the 24 provinces of Ecuador.” Authors must mention how the sample was defined and what percentage of answers were obtained, in relation to the number of questionnaires sent initially.

Line 305 – in the conclusions, the three objectives of this study should be described in a concise and objective way: relationship between psychological distress, work engagement, and the work environment.

The conclusions must describe: the added value of this study, the limitations and future studies.

Author Response

Improvement proposals

Line 83 – “Despite studies on work engagement and the psychological effects of COVID-19 that have been conducted in Latin America, none are available in our immediate environment.” Authors must demonstrate why studies are not readily available.

We wanted to state that no studies about work engagement and the psychological effects of COVID-19 on the Ecuadorian general population have been performed.

New references have been included to show this study gap and stablish comparisons between collectives in Ecuador. This can be seen in the Introduction and the Discussion.

Line 97 – “A non-probabilistic, snowball sample of 2161 questionnaires was obtained from the 24 provinces of Ecuador.” Authors must mention how the sample was defined and what percentage of answers were obtained, in relation to the number of questionnaires sent initially.

The calculation of the sample size has been defined.

The estimated sample size was 2481, with 95% confidence level, 2.2% precision, and a loss adjustment of 20%. Finally, the loss was 12.9%, leaving a sample size of 2161.

As the questionnaires were not sent individually by the investigators, national organisations and participants were asked to share the questionnaire with their colleagues and social networks after completion, through a snow-ball sampling effect. Therefore, it was not possible to know the percentage of answers in relation to the number of questionnaires.

Line 305 – in the conclusions, the three objectives of this study should be described in a concise and objective way: relationship between psychological distress, work engagement, and the work environment.

We have revised the conclusions to show all the relevant outcomes and the key contributions regarding the three main variables.

The conclusions must describe: the added value of this study, the limitations and future studies.

The study limitations have been included in the Discussion section.

The contributions of this study have been presented at the end of the Discussion.